# Mat-O-Covid: Validation of a SARS-CoV-2 Job Exposure Matrix (JEM) Using Data from a National Compensation System for Occupational COVID-19

**DOI:** 10.3390/ijerph19095733

**Published:** 2022-05-08

**Authors:** Alexis Descatha, Grace Sembajwe, Fabien Gilbert, Marc Fadel

**Affiliations:** 1Univ. Angers (University of Angers), CHU Angers, Univ. Rennes, Inserm, EHESP, IRSET (Institut de Recherche en Santé, Environnement et Travail)-UMR_S 1085, IRSET-ESTER, SFR ICAT, CAPTV CDC, F-49000 Angers, France; fabien.gilbert@univ-angers.fr (F.G.); marc.fadel@univ-angers.fr (M.F.); 2Department of Occupational Medicine, Epidemiology and Prevention, Donald and Barbara Zucker School of Medicine, Hosftra University Northwell Health, New York, NY 11021, USA; gsembajwe@northwell.edu

**Keywords:** public health, occupational, COVID-19, SARS-CoV-2, work, job exposure matrix, JEM, compensation, predictivity, validity, accuracy

## Abstract

Background. We aimed to assess the validity of the Mat-O-Covid Job Exposure Matrix (JEM) on SARS-CoV-2 using compensation data from the French National Health Insurance compensation system for occupational-related COVID-19. Methods. Deidentified compensation data for occupational COVID-19 in France were obtained between August 2020 and August 2021. The case acceptance was considered as the reference. Mat-O-Covid is an expert-based French JEM on workplace exposure to SARS-CoV-2. Bi- and multivariable models were used to study the association between the exposure assessed by Mat-O-Covid and the reference, as well as the area under the curve (AUC), sensitivity, specificity, predictive values, and likelihood ratios. Results. In the 1140 cases included, there was a close association between the Mat-O-Covid index and the reference (*p* < 0.0001). The overall predictivity was good, with an AUC of 0.78 and an optimal threshold at 13 per thousand. Using Youden’s J statistic resulted in 0.67 sensitivity and 0.87 specificity. Both positive and negative likelihood ratios were significant: 4.9 [2.4–6.4] and 0.4 [0.3–0.4], respectively. Discussion. It was possible to assess Mat-O-Covid’s validity using data from the national compensation system for occupational COVID-19. Though further studies are needed, Mat-O-Covid exposure assessment appears to be accurate enough to be used in research.

## 1. Introduction

In the context of the COVID-19 pandemic, the workplace seemed to be an important source of exposure, if not the main source of contamination [1,2,3]. However, there have been relatively few comparisons of contamination in different occupations using population-level data [1]. Indeed, assessing workplace exposure of SARS-CoV-2 is possible individually, but is challenging for large population studies. To be able to study work-related SARS-CoV-2, we developed a job exposure matrix (JEM) on SARS-CoV-2: Mat-O-Covid [4,5]. After initially validating reliability among experts and correlation with similar, but not identical, data such as O*Net and CONSTANCES [6], we needed to assess its accuracy against a SARS-CoV-2 workplace exposure assessment. Indeed, there is no standard protocol to quantitatively distinguish between a high agreement between experts and a high agreement with a gold standard. Finding a large and accurate assessment is also difficult, though, as a proxy, we have access to results of COVID compensation.

In France, the health insurance system for work-related disorders and injuries has established a national committee for occupational COVID-19 compensation. In our study, we aimed to validate Mat-O-Covid by comparing the results of the JEM exposure assessment with the conclusions of the first cases of the French COVID committee used as a gold standard.

## 2. Materials and Methods

### 2.1. French Compensation System

The French compensation structure for recognizing the occupational nature of a disease is based on two systems [7]:

A list system (‘Tables’ in French): If the disease is listed as an occupational disease and if the “related conditions” (i.e., diagnosis criteria, time suffering from the condition-delay in diagnosis, sometimes the duration of exposure and type of exposure) are met, the disease is presumed to be occupational and the disease is compensated. A new “Table” was created in 2020 which compensated severe COVID-19 cases for salaried workers in the health and social services sectors. This list system can explain why there is a high recognition rate for COVID-19. 

A complementary system is as follows: If the conditions are not met or if the disease does not appear in the list, compensation may be granted if (1) the victim has a predictable permanent disability rate over 25% and if (2) a committee determines that the disease is directly related to workplace exposure. A national committee was established to study compensation cases of severe COVID-19 for salaried workers in other sectors.

### 2.2. Study

An accuracy study used blinded (deidentified) compensation data for all occupational COVID-19 in France, with the data obtained between end of August 2020 and end of August 2021 (12 months). In the database including all consecutive cases, the sex, age, job title, and acceptance/rejection (and reasons) were available. The job title was coded using the 2008 International Standard Classification of Occupations (ISCO).

The Mat-O-Covid JEM used the ‘Profession et Catégories Socioprofessionnelles PCS 2003’ (French Classification of Occupations) with a metadata crosswalk that transcoded it through ISCO 2008. Mat-O-Covid is an expert-based JEM: a group of four experts in different occupational fields independently coded occupational exposure to SARS-CoV-2 from 0 to 1 [5]. Similarly, three other experts coded prevention methods, allowing the results to be an index associated with the probability of exposure to SARS-CoV-2 “Mat-O-Covid Index” [6].

The main outcome was the compensation results of the national committee, i.e., acceptance or rejection. Only questioning cases for possible of exposure were included and decided by committee as well as the straight acceptance. Rejections for medical discrepancies, time from end of exposure, or medical diagnosis issues were excluded, as well as cases with missing data for job titles.

Standard statistics were calculated using bivariate analyses (Student T-test, Chi²) and multivariable logistic models adjusted for age and sex. Using the compensation results as reference, the sensitivity, specificity, predictive values, likelihood ratio and area under the curve (AUC) of the receiver operative characteristic curve were determined. The optimal threshold was calculated using Youden’s J statistic. A *p*-value lower than 0.05 was considered significant, and the 95% confidence intervals were calculated. The study was included in the Mat-O-Covid project which was approved by the Ethics Committee of Angers Teaching University Hospital (2021-009), Statistical Analysis System v9.4 (SAS Institute Inc., Cary, NC, USA), and Stata V17.0 SE (StataCorp, College Station, TX, USA).

## 3. Results

In the first year of France’s COVID-19 compensation system for salaried employees, 1140 cases were included, with a 95.5% acceptance rate (n = 1089). Available sociodemographic data showed that the population was of senior workers, with a mean age of 55.0 years (±10.0 years, median 57 years, range 23–84 years). There were slightly more men than women: 40% were women (n = 456). The Mat-O-Covid Index was 15.0 per thousand (±10.0, median 18, range 0–35 per thousand).

There was a significant association between the results of the Mat-O-Covid JEM and the Committee’s decision, with a mean Mat-O-Covid index of 15.5 per thousand for an acceptance, versus 4.9 per thousand for a refusal (*p* < 0.0001). After adjusting age and sex, the Mat-O-Covid index odds ratio was 1.12 [1.08–1.17] (*p* < 0.0001). 

The overall predictivity was good with an area under the curve of 0.78 (Figure 1). The optimal threshold was 13 per thousand using Youden’s J statistic, with a sensitivity of 0.67, a specificity of 0.87, a low negative predictive value (0.11) and a high positive predictive value (0.99). Both positive and negative likelihood ratios were significant: 4.9 [2.4–6.4] and 0.4 [0.3–0.4], respectively. The association using 13 per thousand (n = 735, 64.5%) was still significant: the Mat-O-Covid odds ratio was 11.5 [5.1–26.0] (*p* < 0.0001). 

## 4. Discussion

This study further completed the Mat-O-Covid JEM validation by assessing it against field data. There was a strong association and accurate predictivity against compensation insurance results. 

We found a strong association between the Mat-O-Covid JEM and the results of the Committee. The association seemed accurate with a level of AUC at 0.78. The determined threshold (13 per thousand) appears very frequently in the sample with almost two-thirds of subjects reaching it. This is not the only case of a high rate of acceptance in France, there were similar results for musculoskeletal disorders (over 90% of acceptance) [8]. This high acceptance rate has a direct effect on predictive values, but it should not influence sensitivity/specificity and likelihood ratios that are robust against variation of prevalence in the sample. The low negative predictive value of 361 false negatives is an important limitation when using the Mat-O-Covid JEM on an individual level. Although Mat-O-Covid is a research tool for that might also be used for public health purposes, it is for population assessments and is not recommended for individual-level use [6,9]. 

Assessing work-exposure to SARS-CoV-2 has become important for research [10,11]. Different study approaches for this purpose have been developed in other countries based on questionnaire surveys [12]. These surveys are considered more precise in distinguishing task-level differences in exposure among similar job categories, compared to a JEM [13]. However, most large studies do not have specific questions on work, and the JEM would be more accurate when considering asymptomatic cases and estimation of COVID-19, with a lower misclassification bias, very similar to a chemical JEM [14]. An English working group developed a control banding matrix to help employers assess the risks of COVID-19 infection during the pandemic [15], and researchers from Denmark, Netherlands and United Kingdom also developed a JEM defining relevant exposure and workplace characteristics related to exposure to the SARS-CoV-2 [16]. Differences in country and design would be interesting to compare in future studies. 

Some limitations of the JEM approach exist. Only severe cases of COVID-19 can be potentially compensated by the list and complementary system. Public servants and self-employed individuals are not included in the compensation system. However, less severe cases are similar in public servants and self-employed when it comes to COVID-19 exposure. Second, this is the first validation of workplace exposure while concurrently developing the JEM. Although this matrix has been found to correlate with previous studies, including O*Net (US JEM previously created), external validity is lacking, and further studies are needed with different data that factor time, considering circulating virus and variants. Third, the gold standard was expert judgments and not that which was based on direct measures of exposure data. However, members of the Committee were independent, i.e., not involved in the rating of Mat-O-Covid and blinded to the results of Mat-O-Covid. It allowed us to better assess workplace exposure to COVID-19, in more detail than could have been achieved with direct measures of exposure/contamination data.

## 5. Conclusions

In conclusion, Mat-O-Covid appears to provide a fairly accurate assessment of occupational exposure to SARS-CoV-2 that causes COVID-19. Further studies are needed in other workplace and country settings that incorporate direct measures of exposure (real world data), and that span various exposure assessment approaches.

## Figures and Tables

**Figure 1 ijerph-19-05733-f001:**
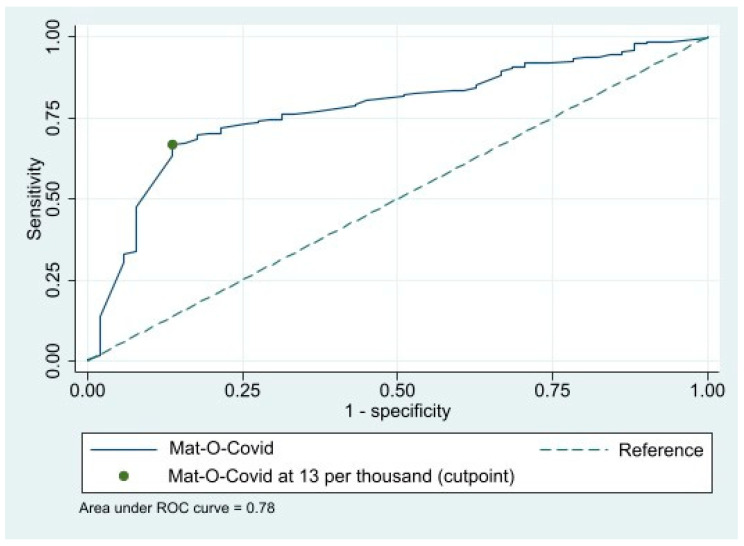
Receiver operative characteristic curve. Mat-O-Covid data were compared to the health insurance decisions on compensation. The Youden’s J statistic illustrated the cut point.

## Data Availability

Data are restricted to French health insurance.

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
