# Peer review of "Mat-O-Covid: Validation of a SARS-CoV-2 Job Exposure Matrix (JEM) Using Data from a National Compensation System for Occupational COVID-19"

_ijerph, 2022, doi:10.3390/ijerph19095733_

Round 1
Reviewer 1 Report
Dear Authors,
I was pleased to read your manuscript. It is overall well written, clear, and straightforward.
Some minor points should be addressed:
Page 1, Abstract, line 44 "multivariate" models are referred to, this means the logistic regression model which is not a multivariate technique. Although the literature on the matter tends to be cumbersome... given the current state of the art I would prefer "multivariable" models instead.
Page 1, Abstract, line 35 Area under the curve (AUC) remove the "s" on "curves".
Page 2, Introduction, line 49: "work" seemed to be an important source of exposure. The semantics of this phrase is strange. I believe that it is not the "work" that is meant is the "workplace" or the "work location". Please find a way to make it more clear.
Page2, 2.2. Study, 1st paragraph, line 77: "between August 2020 and August 2021" is somewhat unspecific. Was it 12 months' data or 13 months' data? I propose adding "(12 months)" or as appropriate to clarify.
Page2, 2.2. Study, 4th paragraph, line 90: "and multivariate logistic models" - as referred above this technique is not multivariate I propose two possible variants for this: 1) "logistic multiple regression models" or 2) "multivariable logistic regression models".
Page 2, 2.2. Study, 4th paragraph, line 94: "and the 95% confidence interval was calculated." it seems to me that "and the 95% confidence intervals were calculated." sounds better.
Page 3 - 3. Results, 2nd paragraph - , line104: "close association"? what is a close association?? maybe " a statistical" or "a significant" association.
Page 3 - 3. Results, Figure 1 - , line114: Figure has a low quality - improve it if possible.
Page 3 - 4. Discussion, 2nd paragraph - , line104: "the association is fair"? what is a fair association?
lines 124-125: "This is not the only case of a high rate of acceptance in France and there was similar results for musculoskeletal disorders (over 90% of acceptance)." maybe "This is not the only case of a high rate of acceptance in France, there were similar results for musculoskeletal disorders (over 90% of acceptance)."
Page 4 - line148: "Second, it is the first validation of workplace exposure while concurrently..." maybe "Second, this is the first validation of workplace exposure while concurrently..."
Page 4 - line 155: "thatn" I think it is meant "than".
Page 4 - 5. conclusions -, line158: "Mat-O-Covid appears to be an accurate assessment.." - I propose: "Mat-O-Covid appears to provide a fairly accurate assessment.." please find a way to make it more close to the level of gathered evidence.
Author Response
I was pleased to read your manuscript. It is overall well written, clear, and straightforward.
Some minor points should be addressed:
Page 1, Abstract, line 44 "multivariate" models are referred to, this means the logistic regression model which is not a multivariate technique. Although the literature on the matter tends to be cumbersome... given the current state of the art I would prefer "multivariable" models instead.
Response. We have mentioned multivariable
Page 1, Abstract, line 35 Area under the curve (AUC) remove the "s" on "curves".
Response. We have mentioned Curve not curves
Page 2, Introduction, line 49: "work" seemed to be an important source of exposure. The semantics of this phrase is strange. I believe that it is not the "work" that is meant is the "workplace" or the "work location". Please find a way to make it more clear.
Response. We have mentioned workplace for more clarity.
Page2, 2.2. Study, 1st paragraph, line 77: "between August 2020 and August 2021" is somewhat unspecific. Was it 12 months' data or 13 months' data? I propose adding "(12 months)" or as appropriate to clarify.
Response. We have mentioned it was end of August of both year (added).
Page2, 2.2. Study, 4th paragraph, line 90: "and multivariate logistic models" - as referred above this technique is not multivariate I propose two possible variants for this: 1) "logistic multiple regression models" or 2) "multivariable logistic regression models".
Response. We have mentioned multivariable logistic regression models
Page 2, 2.2. Study, 4th paragraph, line 94: "and the 95% confidence interval was calculated." it seems to me that "and the 95% confidence intervals were calculated." sounds better.
Response. We have mentioned CI in plural indeed.
Page 3 - 3. Results, 2nd paragraph - , line104: "close association"? what is a close association?? maybe " a statistical" or "a significant" association.
Response. We have mentioned significant.
Page 3 - 3. Results, Figure 1 - , line114: Figure has a low quality - improve it if possible.
Response. We have tried to improved it as a separate figure in the PDF (600 dpi instead of 72)
Page 3 - 4. Discussion, 2nd paragraph - , line104: "the association is fair"? what is a fair association?
Response. We have changed to “seem accurate” since AUC at 0.78 is between fair to good depending on the scale used.
lines 124-125: "This is not the only case of a high rate of acceptance in France and there was similar results for musculoskeletal disorders (over 90% of acceptance)." maybe "This is not the only case of a high rate of acceptance in France, there were similar results for musculoskeletal disorders (over 90% of acceptance)."
Response. We have followed your recommendations.
Page 4 - line148: "Second, it is the first validation of workplace exposure while concurrently..." maybe "Second, this is the first validation of workplace exposure while concurrently..."
Response. We have followed your recommendations
Page 4 - line 155: "thatn" I think it is meant "than".
Response. Sorry, this typo has been corrected.
Page 4 - 5. conclusions -, line158: "Mat-O-Covid appears to be an accurate assessment.." - I propose: "Mat-O-Covid appears to provide a fairly accurate assessment.." please find a way to make it more close to the level of gathered evidence.
Response. Thank you. We have corrected the sentence to reflect your advice: “Mat-O-Covid appears to provide a fairly accurate assessment.”

Reviewer 2 Report
I appreciate the authors' effort to solve an important problem in the area of ​​Covid-19 pandemic but, in my opinion, unfortunately, the final result does not meet the requirements of prestigious scientific journals such as the International Journal of Environmental Research and Public Health.
In particular, the manuscript lacks:
1/ The manuscript lacks a description of the research gap.
2/ The manuscript lacks a clearly formulated aim.
3/ The introduction to the manuscript is very brief and consists of only two short paragraphs. As a result, there is no justification why the research problem raised by the authors is important and the paper deserves to be published.
4/ The results are described very briefly, in a very vague manner. Moreover, some information in this section should be included in the methodology section, e.g. demographics.
5/ Discussion is very laconic and much of this section has practical implications and limitations. However, there is no reference to the literature and other studies of this type.
6/ Conclusions only consist of 2 short sentences. Considering the reputation of IJERPH, this is significantly below the requirements.
7/ The manuscript requires linguistic correction.
In conclusion, after reading the manuscript, unfortunately, I have to admit that it is not mature enough to be published in its current form.
Author Response
Reviewer 2
I appreciate the authors' effort to solve an important problem in the area of ​​Covid-19 pandemic but, in my opinion, unfortunately, the final result does not meet the requirements of prestigious scientific journals such as the International Journal of Environmental Research and Public Health.
Response. We thank the Reviewer for the comment. To explain our position, we had a precise scientific question and we thought a manuscript using the short paper format would be appropriate (slightly less than 1500 words). The discussion is brief mostly because there is a paucity of literature on the topic (16 references, appropriate for a short format). We also are consumers of prestigious journal articles and believe IJERPH is as appropriate for short papers as it is for long ones. We have tried to expound upon the points presented by this Reviewer, while retaining the manuscript as a short paper.
In particular, the manuscript lacks:
1/ The manuscript lacks a description of the research gap. “ Indeed, there is a research gap between a high agreement between experts and a high agreement with a gold standard. Finding a large and accurate assessment is also difficult though we have accessed of results of COVID compensation.”
Response R1. We have have edited the sentence to clarify that there is “no standard protocol to quantitatively distinguish” between high agreement between experts and a high agreement with a gold standard.
2/ The manuscript lacks a clearly formulated aim.
Response R2 We have clarified the aim. “In our study, we aimed to validate Mat-O-Covid by comparing the results of the JEM exposure assessment with the conclusions of the first cases of the French COVID committee used as a gold standard.”
3/ The introduction to the manuscript is very brief and consists of only two short paragraphs. As a result, there is no justification why the research problem raised by the authors is important and the paper deserves to be published.
Response R3. We have added more detail on the research gap based on previous research.
4/ The results are described very briefly, in a very vague manner. Moreover, some information in this section should be included in the methodology section, e.g. demographics.
Response R4 We have clarified the available demographics. We aimed to answer a short question and received the few variables from the database that were also mentioned in the protocol. “Available sociodemographic data showed that the population was of senior workers, with a mean age of 55.0 years (+/- 10.0 years, median 57 years, range 23-84 years). There were slightly more men than women; 40% were women (n=456).”
5/ Discussion is very laconic and much of this section has practical implications and limitations. However, there is no reference to the literature and other studies of this type.
R5. The discussion is short because literature is scarce on this topic. We have expounded in some areas of the manuscript, as the Reviewer advised. We have also mentioned “However, most large studies don’t have specific questions on work, and the JEM would be more accurate when considering asymptomatic cases and estimation of COVID-19, with a lower misclassification bias similar to a chemical JEM.[14] An English working Group developed a control banding matrix to help employers assess the risks of COVID-19 infection during the pandemic, [15] and researchers from Denmark, Netherlands and United Kingdom also developed a JEM defining relevant exposure and workplace characteristics related to exposure to the SARS-CoV2.[16] Differences in country and design would be interesting to compare in future studies.”
6/ Conclusions only consist of 2 short sentences. Considering the reputation of IJERPH, this is significantly below the requirements.
Response R6. We have tried to stay within the short paper format and we hope to have fulfilled IJERPH requirements.
7/ The manuscript requires linguistic correction.
Response R7. We have asked our native English speaking co-authors to review the manuscript for cogency.
In conclusion, after reading the manuscript, unfortunately, I have to admit that it is not mature enough to be published in its current form
Response R8. We have responded to all comments from the Reviewers (including the highly positive ones) and we now hope that you considered mature enough for publication within the short paper format recommended by IJERPH.
Reviewer 3 Report
The article "Mat-O-Covid: Validation of a SARS-CoV-2 Job Exposure Matrix (JEM) using data from a national compensation system for occupational Covid-19" is relevant. However, it is necessary to present the research gap based on previous research. It is also necessary to explain the research methodology in more detail.
Author Response
The article "Mat-O-Covid: Validation of a SARS-CoV-2 Job Exposure Matrix (JEM) using data from a national compensation system for occupational Covid-19" is relevant. However, it is necessary to present the research gap based on previous research. It is also necessary to explain the research methodology in more detail.
Response. We have followed your advice and explained better the rationale, as well the methodology.
“In the context of the Covid-19 pandemic, workplace seemed to be an important source of exposure, if not the main source of contamination.[1–3] However, there have been relatively few comparisons of contamination in different occupations using population-level data.[1] Indeed, assessing workplace exposure of SARS-CoV-2 is possible individually, but is challenging for large population studies. To be able to study work-related SARS-CoV-2, we developed a job exposure matrix (JEM) on SARS-CoV-2: Mat-O-Covid.[4,5] After initially validating reliability among experts and correlation with similar, but not identical, data such as O*Net and CONSTANCES,[6] we needed to assess its accuracy against a SARS-CoV-2 workplace exposure assessment. Indeed, there is no standard protocol to quantitatively distinguish between a high agreement between experts and a high agreement with a gold standard. Finding a large and accurate assessment is also difficult though, as a proxy, we have access to results of COVID compensation. Indeed, there is no standard protocol to quantitatively distinguish between a high agreement between experts and a high agreement with a gold standard. Finding a large and accurate assessment is also difficult though, as a proxy, we have access to results of COVID compensation.”
…
Blinded (deidentified) compensation data for occupational COVID-19 in France were obtained between end of August 2020 to end of August 2021 (12 months). In the database, the sex, age, job title, the acceptance/rejection (and reasons) were available. The job title was coded using the 2008 International Standard Classification of Occupations (ISCO).
…
Standard statistics were calculated using bivariate analyses (Student T-test, Chi²) and multivariable logistic models adjusted for age and sex. Using the compensation results as reference, the sensitivity, specificity, predictive values, likelihood ratio and Area Under the Curve (AUC) of the Receiver Operative Characteristic curve were determined. The optimal threshold was calculated using Youden’s J statistic. A P-value lower than 0.05 was considered significant and the 95% confidence intervals were calculated. The study was included in the Mat-O-Covid project which was approved by the Ethics Committee of Angers Teaching University Hospital (2021-009), Statistical Analysis System v9.4 (SAS Institute Inc., Cary, NC, USA), and Stata V17.0 SE (StataCorp, Texas 77845, USA).
Reviewer 4 Report
Congratulations for your article. It is necessary to recognize coronavirus disease as an occupational disease in certain areas, although on the other hand it is difficult to demonstrate such an association.Reviewing the bibliography, no similar articles have been published, so it is necessary to show the problems and needs that other health sectors may have.
The article is complete and well structured. The title is concise, clear and attractive.
The summary shows the content of the article, with adequate and formal language.
The introduction explains the background and acquaint the reader with the study problem with bibliographic references in an appropriate way. It also clearly formulates the objectives of the study.
The study methodology is correct.
The results are in accordance with the objectives. The discussion and conclusions are adequate. The discussion responds to the stated objectives and compares the results with the most recent and relevant ones published at the moment. It reflects the main limitation of the study while proposing the need to continue studying in this area. The bibliography is adequate, extensive and up-to-date
Author Response
Congratulations for your article. It is necessary to recognize coronavirus disease as an occupational disease in certain areas, although on the other hand it is difficult to demonstrate such an association. Reviewing the bibliography, no similar articles have been published, so it is necessary to show the problems and needs that other health sectors may have.
The article is complete and well structured. The title is concise, clear and attractive.
The summary shows the content of the article, with adequate and formal language.
The introduction explains the background and acquaint the reader with the study problem with bibliographic references in an appropriate way. It also clearly formulates the objectives of the study.
The study methodology is correct.
The results are in accordance with the objectives. The discussion and conclusions are adequate. The discussion responds to the stated objectives and compares the results with the most recent and relevant ones published at the moment. It reflects the main limitation of the study while proposing the need to continue studying in this area. The bibliography is adequate, extensive and up-to-date
Response. We thank the reviewer for his time and a such positive review.
Round 2
Reviewer 2 Report
I appreciate the authors' effort to improve the manuscript according to the comments. The authors' explanations and all corrections made the article more refined now. I am glad that the authors used the comments they received to improve the quality of the manuscript.
I confirm that the new version of the manuscript contains corrections in the following areas:
1/ The aim of the article has been clarified.
2/ More details have been added on the research gap based on previous research.
3 / The available demographics have been clarified.
4 / Some parts of the discussion section have been expounded.
5 / The manuscript has been linguistically corrected.
As the authors responded to all comments and corrected the article as required, I suggest to accept the manuscript in its present form.
Author Response
Thank you very much for your time.
Reviewer 3 Report
The paper "Mat-O-Covid: Validation of a SARS-CoV-2 Job Exposure Matrix (JEM) using data from a national compensation system for occupational Covid-19" is relevant.
I requested the corrections, but they were not made to the manuscript. There is no clarity from the research on the subject and the research gap. In addition, the research method does not allow for replicability, due to the lack of step by step.
Author Response
We have tried to do our best to answer to reviewer 3 concerns though not clear enough, as well as the other three. Corrections are in green and manuscript with correction are in supplemental material (as in the last version). We have added a complete answer enclosed.
